# Conditioned Medium of Intervertebral Disc Cells Inhibits Osteo-Genesis on Autologous Bone-Marrow-Derived Mesenchymal Stromal Cells and Osteoblasts

**DOI:** 10.3390/biomedicines12020376

**Published:** 2024-02-06

**Authors:** Shuimu Chen, Andreas S. Croft, Sebastian Bigdon, Christoph E. Albers, Zhen Li, Benjamin Gantenbein

**Affiliations:** 1Tissue Engineering for Orthopedics & Mechanobiology (TOM), Bone & Joint Program, Department for BioMedical Research (DBMR), Faculty of Medicine, University of Bern, 3008 Bern, Switzerland; shuimu.chen@unibe.ch (S.C.);; 2Graduate School for Cellular and Biomedical Sciences (GCB), University of Bern, 3012 Bern, Switzerland; 3Department of Orthopedic Surgery & Traumatology, Inselspital, University of Bern, 3010 Bern, Switzerland; sebastian.bigdon@insel.ch (S.B.); christoph.albers@insel.ch (C.E.A.); 4AO Research Institute Davos, 7270 Davos, Switzerland; zhen.li@aofoundation.org

**Keywords:** intervertebral disc, conditioned medium, mesenchymal stromal cell, osteoblast, osteogenesis, spinal fusion

## Abstract

Low back pain (LBP) is associated with the degeneration of human intervertebral discs (IVDs). Despite progress in the treatment of LBP through spinal fusion, some cases still end in non-fusion after the removal of the affected IVD tissue. In this study, we investigated the hypothesis that the remaining IVD cells secrete BMP inhibitors that are sufficient to inhibit osteogenesis in autologous osteoblasts (OBs) and bone marrow mesenchymal stem cells (MSCs). A conditioned medium (CM) from primary human IVD cells in 3D alginate culture was co-cultured with seven donor-matched OB and MSCs. After ten days, osteogenesis was quantified at the transcript level using qPCR to measure the expression of bone-related genes and BMP antagonists, and at the protein level by alkaline phosphatase (ALP) activity. Additionally, cells were evaluated histologically using alizarin red (ALZR) staining on Day 21. For judging ALP activity and osteogenesis, the Noggin expression in samples was investigated to uncover the potential causes. The results after culture with the CM showed significantly decreased ALP activity and the inhibition of the calcium deposit formation in alizarin red staining. Interestingly, no significant changes were found among most bone-related genes and BMP antagonists in OBs and MSCs. Noteworthy, Noggin was relatively expressed higher in human IVD cells than in autologous OBs or MSCs (relative to autologous OB, the average fold change was in 6.9, 10.0, and 6.3 in AFC, CEPC, and NPC, respectively; and relative to autologous MSC, the average fold change was 2.3, 3.4, and 3.2, in AFC, CEPC, and NPC, respectively). The upregulation of Noggin in residual human IVDs could potentially inhibit the osteogenesis of autologous OB and MSC, thus inhibiting the postoperative spinal fusion after discectomy surgery.

## 1. Introduction

Low back pain (LBP), a significant global health issue and complex condition, arises from a multifaceted interplay of factors [1]. Reverberating across societies worldwide, the challenges posed by LBP are profound: reduced quality of life, impaired mobility, and substantial healthcare costs [2]. The frequent cause is intervertebral disc (IVD) degeneration [3,4]. 

Intervertebral discs are pivotal components of the vertebral column, are situated between adjacent vertebrae, and play a crucial role in maintaining spinal function. These discs possess a distinct structure and composition that enable their essential functions. Comprising the outer annulus fibrosus (AF) and the inner nucleus pulposus (NP), intervertebral discs are designed to withstand mechanical stresses while providing flexibility and shock absorption to the spine [5]. Cartilaginous endplates (CEPs) connect the disc to the adjacent vertebral bodies, enabling nutrient exchange and maintaining disc health [4]. Intervertebral disc degeneration (IDD) is closely associated with the development of LBP. Notably, the degradation of the NP’s water-binding capacity contributes to a diminished disc height and impaired load distribution, which can lead to nerve irritation and pain [6]. Understanding the structure and function of the IVD is fundamental for comprehending spinal health and addressing various spinal disorders. The vertebral bodies on the upper and lower sides of the IVD contain osteoblasts (OBs) and bone marrow. Mesenchymal stem cells (MSCs) derived from the bone marrow can differentiate into osteoblasts. After discectomy in spinal surgery, these MSCs play an important role in the fusion of the spine, contributing to bone formation and repair [7].

The current surgical approach to addressing human IVD degeneration often involves the removal of the affected tissue, followed by the placement of a cage to replace the primary space and promote spinal fusion [4]. However, despite these efforts, there exists a persistent challenge in achieving successful fusion outcomes [8]. Non-fusion or failed fusion after surgery is a recurring concern [9], and identifying the underlying factors has been a focal point of research. One possible explanation is linked to the intricate role of bone morphogenetic protein (BMP) antagonists secreted by the IVD itself [10,11]. These antagonists have been associated with inhibiting osteogenesis in the presence of IVD cells [12,13]. Furthermore, co-culture experiments differentiating bone marrow-derived MSC in monolayer to OBs in close contact with IVD tissue revealed obvious zones of inhibition as shown with alizarin red (ALZR) staining (see Figure 2 in ref. [14]). The appearance of inhibition was similar to the agar diffusion methods of testing antibiotics in bacteria [15]. However, in previous studies, all evidence was solely collected from allogenic donors, which can impact the degree of inhibition or could be overlaid by other paracrine effects [12,13]. There is a lack of research on the current understanding of how various cell sources of the human IVD interplay with autologous OBs or MSCs, which are all involved in spinal fusion after discectomy.

In cell biology and tissue engineering, a medium collected from cells is known as the “conditioned medium (CM)” or “secretome”, and it has gained significant attention in recent years due to its assumed regenerative potential. A CM refers to a cell culture medium that has been in contact with primary cells for a specific duration. This medium becomes enriched with various soluble secreted factors, including growth factors, cytokines, and extracellular vesicles, which are released by the cultured cells. These soluble factors can profoundly influence the behavior of cells, including their proliferation and differentiation [16]. Several studies have revealed its role in promoting tissue regeneration [17,18]. Additionally, researchers have also found that an animal-derived nucleus pulposus cell CM can stimulate heterologous MSC differentiation [19]. However, there are no literature reports on the effects of a human IVD-derived CM on autologous surrounding cells, such as OBs and MSCs. 

This current study aimed to investigate the relationship between the CM of human IVD and donor-matched OBs/MSCs, exploring how these secreted factors present in the CM influence the cellular behavior of autologous OBs/MSCs. Furthermore, new treatment strategies and therapeutic guidelines could be identified to improve spinal fusion. 

## 2. Materials and Methods

### 2.1. Human Materials and Cell Isolation

Multiple samples, i.e., bone fragments, intervertebral disc tissues, and human bone marrow aspirates were collected from the same patients undergoing spinal surgery at the Insel University Hospital with written consent (Table 1 and Table 2). Seven donor-derived MSC/OB cells were used for the culture in the CM obtained from autologous IVD and nine donors for the comparison of Noggin expression between IVD and autologous MSC/OB. All the donor tissues and cells were anonymously collected with written consent. The approvals were either collected under the Swissethics approval (#2019-00097) or were obtained under the general consent of the Insel University Hospital, which also covers the anonymization of health-related data and biological material. Bone fragments from injured vertebral body were cut into smaller pieces of 3–5 mm in diameter. The pieces were then washed with phosphate-buffered saline (PBS). Afterwards, the washed bone pieces were transferred to T75 flasks for culture with α-MEM medium (10% fetal bovine serum, FBS, Sigma-Aldrich, St. Louis, MO, USA; 1% penicillin/streptomycin, P/S, Sigma-Aldrich, Inc., Buchs, Switzerland). Primary OBs were expanded using the active outgrowth technique and selected for plastic adherence: the medium was exchanged after ~one week of culture time and when several starting populations of OBs were observed. IVD tissues were processed within 24 h after surgery and divided into AF, CEP, and NP tissues (usually performed in the operation theatre by an experienced spine surgeon). Then, these tissues were sequentially digested with 1.9 mg/mL pronase (Roche, Basel, Switzerland) for one hour and collagenase II (129 U/mL for AF, 64 U/mL for NP, and 1560 U/mL for CEP; Worthington, London, UK) on a plate shaker at 37 °C overnight. The digested tissue mixture was then passed through a 100μm cell strainer (Falcon, Becton Dickinson, Allschwil, Switzerland) to filter out residual tissue fragments. The obtained cells (AFC, CEPC, and NPC) were cultured in the low-glucose (1 g/L) Dulbecco’s Modified Eagle Medium (LG-DMEM, Gibco, Life Technologies, Zug, Switzerland; including 10% FBS, 1% P/S). MSCs were isolated from the mononuclear cell fraction obtained through gradient centrifugation (Histopaque-1077, Sigma-Aldrich) of bone-marrow samples aspirated from vertebrae during spinal surgery (~5–10 mL) [20]. The MSCs were then expanded in an alpha minimum essential medium (α-MEM, from Sigma-Aldrich) containing 10% FBS, 1% P/S, and 2.5 ng/mL basic fibroblast growth factor 2 (bFGF2, Peprotech, London, UK) [21]. The cellular phenotypes were verified after expansion through the expression of key marker genes, such as the ratio of relative gene expression between collagen type 2 alpha 1 chain (COL2A1) and collagen type I alpha 2 chain (COL1A2), i.e., col2/col1 ratio. The difference for all cell types is attached in Appendix A.

### 2.2. Alginate Bead and Conditioned Medium

For the generation of the CM, human OB, MSC, and IVD cells (AFC, CEPC, and NPC; passage 2–4) were encapsulated in 1.2% alginate (Fluka, Basel, Switzerland) dissolved in 0.9% sodium chloride solution at a density of 4 million/mL. Then, the alginate solution was flown at a constant rate at 1.5 mL/min (syringe pump TI—Part # 78-8100—Model No. 100, KD Scientific, Holliston, MA, USA) through a 22G needle and dropped into a 102 nM CaCl_2_ salt solution, which immediately solidified and generated the beads. Within 5 min, the beads underwent rinsing with a 0.9% sodium chloride solution before being transferred to 6-well plates (30 beads/well) and cultured with 4 ml α-MEM (10% FBS, 1% P/S). After three days, the CM was then collected and filtered with a 0.22 μm filter and stored at 4 °C prior to further use (Figure 1). 

### 2.3. Cell culture with Conditioned Medium

Human OB/MSC cells were seeded at 2 × 10^4^ cells/well in 12-well plates in the basal medium (α-MEM; 10% FBS, 1% P/S) and left overnight for cell adherence. Osteogenic medium (α-MEM containing 10% FBS, 1% P/S, 0.1 mM L-ascorbic acid-2-phosphate, 20 mM beta-glycerophosphate, and 200 nM Dexamethasone, from Sigma-Aldrich), together with CM from corresponding cells, was added into each well (2 mL, ratio 1:1) of the positive control, AFC group, CEPC group, and NPC group. The negative control was cultured in the basal medium without additional factors. The medium was refreshed twice a week (Figure 2).

### 2.4. RNA Extraction and Relative Gene Expression Using qPCR

Total RNA was extracted from human OB/MSC cells cultured with donor-matched IVDs’ conditioned medium on Day 10, evaluating bone-related genes expression, including alkaline phosphatase (*ALP*), runt-related transcription factor 2 (*RUNX2*), osteopontin (official name is secreted phosphoprotein 1, *SPP1*), osterix (official name is transcription factor SP7, *SP7*), osteocalcin (official name is bone gamma-carboxyglutamic acid-containing protein, *BGLAP*), and *COL1A2*. The BMP antagonists (*Noggin*, *Chordin*, and *Gremlin1*) were also measured at this time point. As a marker of cell phenotype of primary cells, the gene expression of *COL2A1* was also investigated, which should be high in NPC and CEPC but should be absent in OB and low or absent in MSC [22]. The relative gene expression was normalized to the positive control. Cells from the same donor were collected for RNA extraction to compare the expression of Noggin between human OB/MSC and IVD, and the result in IVD was relative to the control (OB/MSC). The isolated RNA underwent conversion to cDNA using the High-Capacity cDNA Reverse Transcription kit (#4368814; Thermo Fisher Scientific, Inc., Reinach, Switzerland). The cDNA was then mixed with iTaq Universal SYBR Green Supermix (#1725122; Bio-Rad Inc., Cressier, Switzerland) and human-specific oligonucleotide primers sourced from Table 3. Subsequently, quantitative polymerase chain reaction analysis was conducted using the CFX96™ Real-Time System (#185-5096; Bio-Rad Inc., Cressier, Switzerland). Relative gene expression was analyzed by the application of the threshold cycle using the 2^−ΔΔCt^ method [23]. The ribosomal 18S RNA and GAPDH genes were used as reference genes.

### 2.5. Alkaline Phosphatase Activity

After ten days of osteogenic differentiation, the medium in the human OB/MSC was removed, and cell layers were washed with tris-buffered saline (TBS). CelLytic (Sigma-Aldrich) was then used for cell lysis, followed by two rounds of 15s of ultrasonication for each sample. Then, they were centrifuged at 500× *g* for 10 min at 4 °C to obtain the supernatant. The ALP activity was quantified using a phosphatase assay kit (Sigma-Aldrich), and absorbance was measured under 405 nm by using a microplate reader (SpectraMax M5, Bucher Biotec, Basel, Switzerland).

### 2.6. Histology

Alizarin red staining was utilized to detect the mineralized matrix in human OBs/MSCs after osteogenic differentiation. Initial fixation of cells was conducted overnight using 4% formaldehyde, followed by two rinses with distilled water. Subsequently, all wells were exposed to a 2% alizarin red solution (Sigma-Aldrich) for 45 min. Finally, the solution was aspirated, and the residual dye was washed away with PBS before capturing microscopic images. 

### 2.7. Statistics

The results of relative gene expression and ALP activity were evaluated using one-way ANOVA with Tukey’s multiple comparisons test. All experiments were performed in triplicates. The statistical analyses were conducted using GraphPad Prism 6.0 (GraphPad Software, San Diego, CA, USA) software, and a *p* value < 0.05 was considered statistically significant. N = number of biological replicates is indicated in all graphs.

## 3. Results

### 3.1. Effects of CM from Human IVD on Autologous OB and MSC Differentiation

After 10 days of culture, ALP activity was measured in MSC (Figure 3A). In the positive control, ALP activity was found to be 193.90 ± 60.65 mU/mL. This was significantly higher than the negative control (13.36 ± 8.40 mU/mL, *p* < 0.0001). The culture of MSC with the CM from CEPC or NPC resulted in a significant decrease in ALP activity (*p* = 0.0056 and *p* = 0.0162, respectively). However, treatment with the CM from AFC did not have inhibitory effects on ALP activity compared to that of the positive control (*p* = 0.9790). Similarly, ALP activity was decreased in OBs that were treated with the CM from IVD cells (Figure 3C). In addition, compared with the AFC and CEPC groups, the decrease in ALP activity in the NPC group was the most significant (*p* = 0.0248 for AFC group, *p* = 0.0022 for CEPC group, and *p* < 0.0001 for NPC group compared with the positive control). Much of the ALP activity of human MSCs or OBs was suppressed when the CM from the donor-matched IVD was added to the culture. Alizarin red staining on Day 21 showed less calcium deposit in MSCs and OBs treated with the donor-matched IVD CM (Figure 3B,D). This result was congruent with the result of the ALP assay on Day 10. Both results from the human MSCs and OBs revealed that their osteogenesis was negatively affected by secreted molecules from autologous IVD. 

### 3.2. Gene Expression in Human MSCs and OBs upon Stimulation with CM

To investigate the impact of the CM derived from the autologous IVD on the expression of bone-related genes (*ALP*, *RUNX2*, *BGLAP*, *SP7*, *SPP1*, *COL1A2*) and BMP antagonists (*Noggin*, *Gremlin1*, *Chordin*) on Day 10, we performed qPCR analysis on experimental groups and control groups, and the results were presented as fold changes compared to the positive control. Under the influence of the CM, alterations in gene expression profiles were observed in MSCs. Notably, *ALP*, *RUNX2*, and *SP7* maintained varying degrees of reduced expression levels (Figure 4A,B,D), signifying their response to the applied perturbation. Among them, when the CEPC CM was added, the expression levels of the above three genes were reduced. In contrast, after adding the AFC CM, only the expression of *ALP* decreased (Figure 4A). Similarly, only the expression of *RUNX2* was observed to be inhibited when the NPC CM was added (Figure 4B). These observations suggest that *ALP*, *RUNX2*, and *SP7* may be particularly influenced by factors associated with the CM derived from the autologous IVD. However, the gene expression of other bone-related genes, like *BGLAP*, *SPP1*, and *COL1A2*, distinct from *ALP*, *RUNX2*, and *SP7*, was not affected by the CM (Figure 4C,E,F). No significant reduction in all of them was found compared to the positive control, indicating a potential resistance to the interference represented by the CM. To assess the expression of BMP antagonists of human MSCs in response to the autologous CM from IVD, we examined the gene expression profiles of three BMP antagonists: *Noggin*, *Gremlin1*, and *Chordin*. Surprisingly, our analysis revealed that the gene expression levels of all three BMP antagonists remained unaltered in the presence of the CM, as compared to the positive control (Figure 4G–I). Compared to MSCs, OBs also exhibited almost similar results in the gene expression of bone-related genes and BMP antagonists when cultured with the CM derived from the autologous IVD. The stimulation of the human primary OBs with the CM did not decrease the gene expression of bone-related genes after 10 days (Figure 5A–F). The same result could also be observed in BMP antagonists, and no significant differences were found compared to the positive control (Figure 5G–I). In summary, there was a decrease in gene expression for a subset of the tested genes when CM derived from the autologous IVD was added, as compared to the positive control. However, most of the results showed no significant change for ten days’ culture with CM, either in human primary MSCs or OBs.

### 3.3. Noggin Expression in Human Primary OB, MSC, and IVD from Same Patient

To gain additional insight into the outcome of the ALP assay and alizarin red staining, we conducted qPCR of human MSCs, OBs, as well as the three predominant cell types within the donor-matched IVD, namely, AFC, CEPC, and NPC, and we elucidate the inhibitory effect on the osteogenesis of human primary MSC/OB following the culture with the CM derived from the autologous IVD. Subsequently, the relative expression levels of *Noggin*, one of important BMP antagonists, was quantified. The result showed that the average expression of *Noggin* in IVD was mainly increased compared to the autologous human primary MSCs (Figure 6A). However, among the tested AFC from five donors, the expression level of *Noggin* in three donors was relatively lower than that of MSC. Congruent results were found in the NPC group. There were two donors below baseline and the expression level of *Noggin* in the MSC. In contrast, a little different from the result of *Noggin* in the MSC and the corresponding IVD, a strong expression of *Noggin* was observed in all cell types of the human IVD relative to the autologous OB, including all the tested donors (Figure 6B). These results reveal that the average expression of *Noggin* was upregulated in the human primary IVD relative to autologous MSCs or OBs.

## 4. Discussion

As a common health problem, LBP treatment involves a range of strategies, from conservative management to surgical interventions [24,25,26,27]. However, it is important to acknowledge the potential issues associated with surgical approaches because of non-fusion in up to 30% of the cases [28]. The efficacy of some surgical procedures for LBP remains a topic of debate [29]. Hence, finding the possible cause to solve the problem of non-union is necessary.

Li et al. [30] conducted a systematic review and meta-analysis on the impact of smoking on fusion rate in spinal fusion surgery. Smoking was considered an important factor and led to a poor outcome. Moreover, patients with osteoporosis are at a greater risk of non-fusion due to compromised bone quality and reduced bone density [31]. Several articles also discovered that Diabetes mellitus could affect bone healing because insulin is considered a key molecule in bone metabolism and growth after spinal fusion [32,33]. Importantly, the residual human IVD tissue that was possibly left behind after the removal of the IVD, or even the fact that the “IVD-niche” is absolutely not osteoinductive, might affect the outcome and successful ossification of spinal fusion [13]. Previous research by our group showed that the human IVD was potentially associated with inhibiting the osteogenesis of primary osteoblasts in a porous-membrane-separated co-culture system [11]. These studies provide insights into the impact of retained IVD tissue or cells. As for how residual disc tissue affects spinal fusion, the mechanism is not fully understood.

We collected postoperative human IVD tissue, bone fragments, and bone marrow from the same patients and found that the CM derived from IVD cells could inhibit the ALP activity and calcium deposit formation of autologous OBs at the protein level and histology (Figure 3C,D). It means residual disc tissue may secrete a cocktail of molecules into the CM, including numerous soluble factors and extracellular vesicles, and they suppress the formation of a new bone at the original disc space after spinal surgery. From the results in Figure 3, all main cell types of the human IVD have a similar effect on autologous OBs by the CM. We can speculate that these molecules that inhibit OBs are secreted into the CM by each cell type. Interestingly, the human bone-marrow-derived MSC, potentially differentiating into OBs, was measured, and exhibited similar results as the OBs. This indicates that these molecules from the CM also show an inhibition on the osteogenesis of MSCs. Recent studies have shown that the molecules naturally released from cells have a lot of capabilities: anti-inflammation or pro-inflammation, pro-angiogenesis, regeneration, and tissue repair [16,34,35,36]. Yang et al. found that murine macrophages RAW 264.7 cultured in the CM of Rat AFC or NPC increased the levels of inflammatory cytokines [34]. This confirms that the CM derived from IVDs can indeed affect the biological behavior of other cells. In our study, both the human OBs and MSCs were observed not to follow an expected behavior for osteogenesis when cultured with the CM derived from an autologous IVD. This is a little different from the present data about the effect of the CM on the IVD.

Usually, bone-related genes are important indicators of ossification. They reflect the activity of cells in promoting osteogenic differentiation to a certain extent [37,38,39]. After culturing MSCs for ten days using the CM derived from different cell types of an autologous human IVD, we found that in MSC, except for the expression of individual bone-related genes that was inhibited, the expression of most bone-related genes remained stable. The result does not match the previous result of ALP assay as well as alizarin red staining (Figure 3). We think that it may be due to insufficient cultivation time. Some bone-related genes need to be measured at an intermediate or late stage [40,41,42,43]. As for the expression in *ALP*, *RUNX2*, and *SP7*, it is unclear why only some groups have changes in gene expression and some do not, possibly because of limited donors (*n* = 3). Similarly, the result of human OBs in the bone-related genes expression is almost consistent with that of MSCs, with no significant changes recorded on Day 10. This suggests that residual disc tissue does not have enough negative impact on postoperative spinal fusion in the early stages. Importantly, a sufficient recovery time needs to be given when evaluating the effect of intervertebral fusion after spinal surgery. Early relevant indicators may not fully reflect the long-term postoperative healing. Relevant evidence can also be obtained from the expression of BMP antagonists in human MSCs and OBs (Figure 4G–I and Figure 5G–I). As important regulators of bone formation, the BMP antagonists can negatively regulate the downstream signaling of the BMP pathway by binding to BMP2, thereby inhibiting osteogenic differentiation [44].

In this study, we observed notably elevated levels of *Noggin* expression in three cell types of the human IVD (AFC, CEPC, and NPC) when compared to their autologous counterparts, namely, MSCs and OBs. This has a positive correlation with the inhibitory effect of the CM on osteogenesis. This differential expression of *Noggin* across these cell types holds significant implications and warrants further exploration. Noggin is a well-established antagonist of BMP, which is crucial for signaling molecules involved in the regulation of various cellular processes, including differentiation and tissue development [44]. The higher expression of *Noggin* in AFC, CEPC, and NPC may indicate a potential mechanism that Noggin is secreted into the CM by these cell types and inhibits BMP signaling by binding directly to BMP-2, preventing it from interacting with its cell surface receptors [45,46]. This interference blocks the downstream signaling cascade that leads autologous MSCs and OBs to osteogenic differentiation, thereby exerting control over their differentiation and functional properties. The findings demonstrate a specific regulatory role of Noggin derived from IVD cells in the cellular microenvironments nearby. Thus, the high expression level of Noggin in human IVD cells may be a potential inhibitor of autologous OBs and MSCs.

However, our research has some limitations. Noggin concentration in CM was not measured at the protein level this time. Proteomics via mass spectrometry of the CM could be performed in the future to investigate the precise and possible influence from other components in the CM on the experimental results. Furthermore, future research should concentrate on the specific composition of the IVD-derived CM. Thus, mass spectrometry of the proteome might shed new light into the factors that may be important here. Additionally, previous experiments have demonstrated the interplay between BMP2 stimulation and Noggin inhibitor in a dose-dependent manner [47]. Specific inhibition of the most important antagonists, like Noggin, Gremlin-1, or Chordin, by specific antibodies and a down-stream analysis of the SMAD signaling through Western blotting would provide key insights into the mechanism. Finally, the isolation of specific extracellular vesicles (EVs) from the CM of IVD cells might be of therapeutic value as an anti-osteoblastic drug where osteogenesis should be prevented such as in ectopic bone formation, for instance, in diffuse idiopathic skeletal hyperostosis (DISH) [48]. 

Finally, our current cell culture results confirmed previous evidence of high expression of BMP inhibitors in IVD cells. Thus, achieving better spinal fusion antagonists of inhibitors is also of a high interest. Here, L51P, a BMP2 analog with a leucine substitution at position 51 by proline, a molecule that binds to BMP inhibitors, is of high therapeutic value [49]. Recently, the effectiveness of three different ratios of BMP2/L51P mixtures was confirmed in an elderly rat animal study, and these mixtures clearly demonstrated strong ossification in a discectomy/spinal fusion model [50].

## 5. Conclusions

Our study revealed a distinctive difference in high *Noggin* expression in human AFC, CEPC, and NPC relative to autologous OBs and MSCs. These findings also demonstrated the involvement of human IVD cells-derived molecules, including Noggin, potentially in the regulation of autologous OBs and MSCs and the inhibition of osteogenesis in these cells through paracrine signaling. It offers an insight into the process of the osteogenesis and knockdown of *Noggin* in human IVD cells, possibly improving the postoperative spinal fusion. While these findings present promising avenues for therapeutic interventions, further research is needed to fully understand the potential regulatory mechanism. Importantly, the complete removal of the affected IVD during spinal surgery for LBP can effectively block the impact of IVD on osteogenesis, which plays a crucial role in postoperative intervertebral fusion.

## Figures and Tables

**Figure 1 biomedicines-12-00376-f001:**
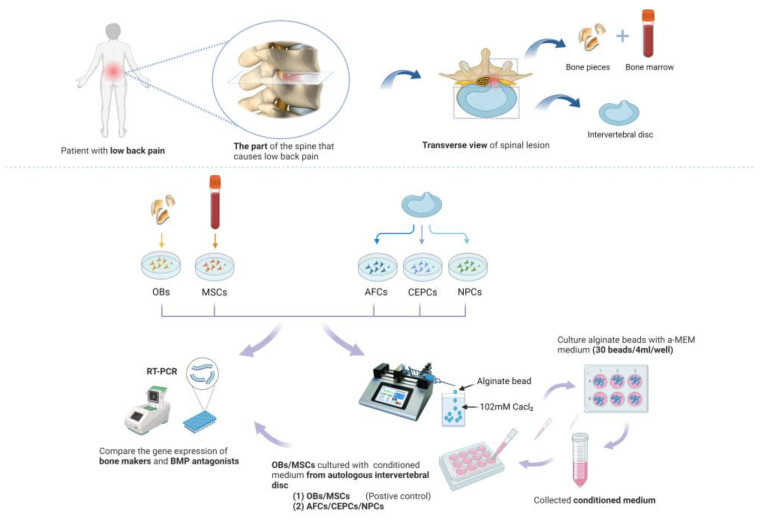
Experimental design for the human OB or MSC cultured with CM from autologous IVD cells. Bone fragments, bone marrow, and IVD tissue samples from spinal surgery were separated to obtain human OBs, MSCs, and the main three types of cells in IVD, including AFC, CEPC, and NPC. To produce the CM of IVD cells, the AFC, CEPC, and NPC were embedded in 3D alginate beads and cultured in the medium. CM from different cell types was collected every three days and added into the autologous OB or MSC in 12-well plate.

**Figure 2 biomedicines-12-00376-f002:**
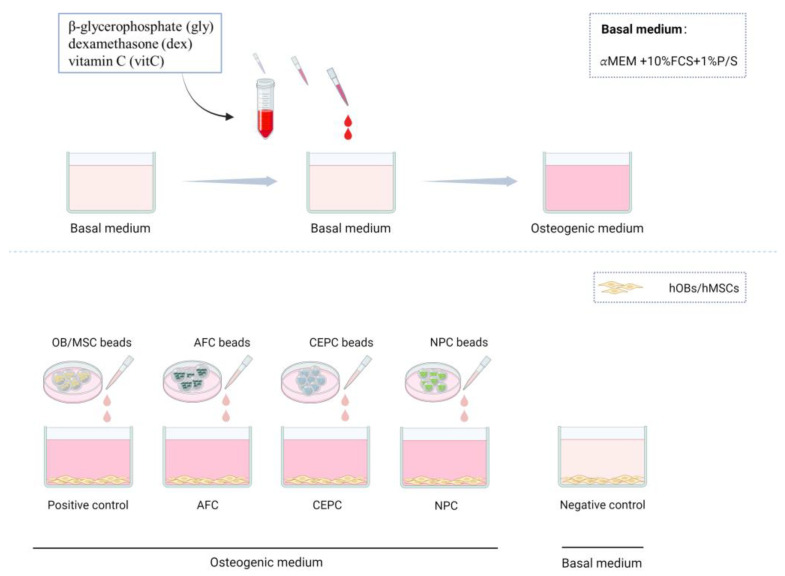
All the five experimental groups were cultured in osteogenic medium except for the negative control. The group added the conditioned medium from human OBs or MSCs was regarded as positive control.

**Figure 3 biomedicines-12-00376-f003:**
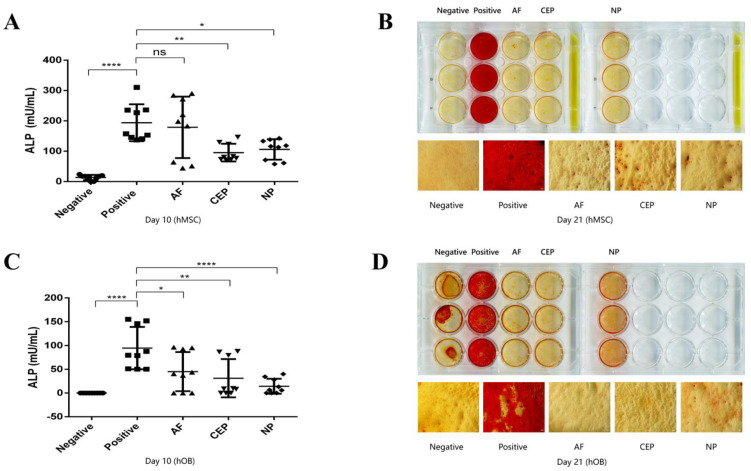
The osteogenesis of human OB/MSC was inhibited after culture with CM from autologous IVD cells. (**A**,**C**) ALP assay showed ALP activity significantly decreased when adding CM from IVD cells into autologous OB and MSC to culture for 10 days. (**B**,**D**) A trend of lower calcium deposit could be observed on Day 21 using alizarin red staining after OBs and MSCs were cultured with CM from different autologous IVD cells. (Calcium deposits: Bright orange red; Scale bar, 100 µm. Mean ± SD. *p*-value, ns > 0.05; * < 0.05; ** < 0.01; **** < 0.0001; *n* = 3–5.)

**Figure 4 biomedicines-12-00376-f004:**
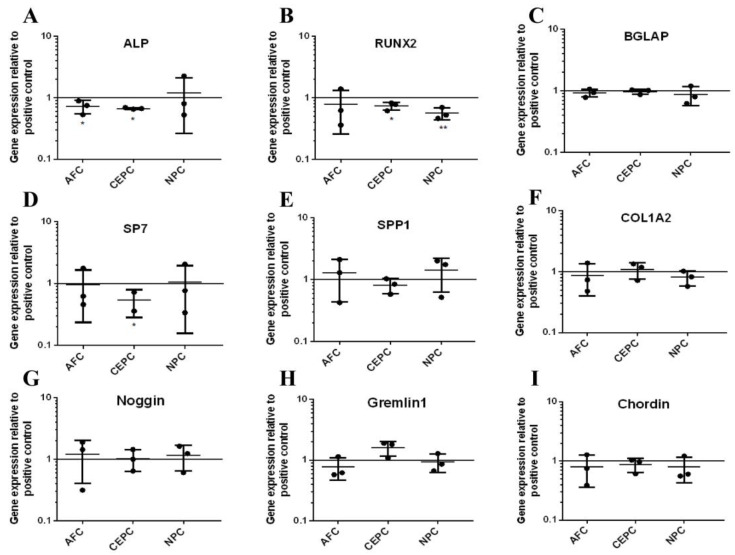
The relative gene expression of bone-related genes and BMP antagonists after human MSC cultured with CM from autologous IVD cells on Day 10. (**A**–**F**) The relative gene expression of bone-related genes was significantly downregulated after adding CM into the autologous MSC but mostly remained unchanged. (**G**–**I**) Relative gene expression of BMP antagonists when cultured with CM for ten days, and no significant changes were noticed. (Mean ± SD, the data were normalized to positive control, *p*-value, * < 0.05; ** < 0.01; *n* = 3–5.)

**Figure 5 biomedicines-12-00376-f005:**
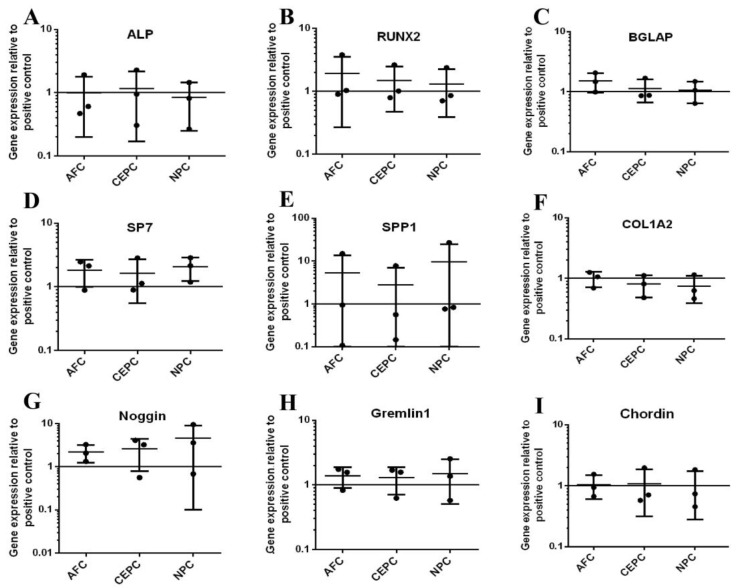
The relative gene expression of bone-related genes and BMP antagonists after human OB cultured with CM from autologous IVD cells on Day 10. (**A**–**F**) The relative gene expression of bone-related genes in OB had no change after adding CM inside. (**G**–**I**) Compared to the positive control, the gene expression of BMP antagonists in experimental groups, AFC, CEPC, and NPC groups, was relatively stable. (Mean ± SD, the data were normalized to positive control, *p*-value; *n* = 3–5.)

**Figure 6 biomedicines-12-00376-f006:**
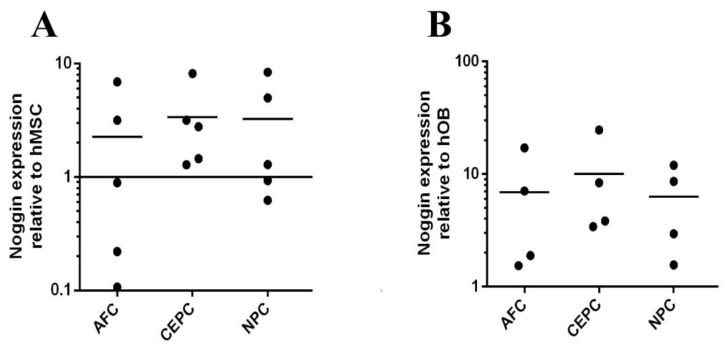
The average expression of *Noggin* was mainly higher in all cell types of human IVD compared to autologous OB, as well as MSC. (**A**) Compared to autologous MSC, the expression of *Noggin* in human IVD increased. (**B**) The same result could be observed significantly between human IVD and corresponding OB. (For MSC and OB, shown are the means, the data were normalized to MSC/OB group, *n* = 3–5).

**Table 1 biomedicines-12-00376-t001:** The details of human donors for culture with conditioned medium from autologous IVD.

Donor	Patient Birth	Sex	Type	IVD Tissues	Level	Bone Marrow	Bone Fragments
1	1988	F	T	√	T12-L1	x	√
2	1988	M	T	√	L1-L2	x	√
3	1950	F	T	√	L1-L2	x	√
4	1985	F	T	√	T12-L1	x	√
5	1946	M	-	√	L3-L4	√	x
6	1961	M	-	√	L3-L4	√	x
7	1988	M	T	√	L1-L2	√	x

Donor’s list of IVD tissues, bone marrow, and bone fragments for the culture experiments with conditioned medium. F, female; M, male; T, trauma; -, missing information; √, collected; x, not collected.

**Table 2 biomedicines-12-00376-t002:** The details of donors for comparison of Noggin expression between IVD and autologous MSCs/OBs.

Donor	Patient Birth	Sex	Type	IVD Tissues	Level	Bone Marrow	Bone Fragments
1	1988	M	T	√	L1-L2	x	√
2	1950	F	T	√	L1-L2	x	√
3	1985	F	T	√	T12-L1	x	√
4	1947	F	T	√	L1-L2	x	√
5	1946	M	-	√	L3-L4	√	x
6	1961	M	-	√	L3-L4	√	x
7	1988	M	T	√	L1-L2	√	x
8	1994	M	T	√	T12-L1	√	x
9	1947	F	T	√	L1-L2	√	x

F, female; M, male; T, trauma; -, missing information; √, collected; x, not collected.

**Table 3 biomedicines-12-00376-t003:** Human gene primers for qPCR.

Gene	Accession No.	Forward Sequence	Reverse Sequence
*18S*	NR_145820.1	CGA TGC GGC GG C GTT ATT C	TCT GTC AAT CCT G TC CGT GTC C
*GAPDH*	NM_001289745.2	ATC TTC CAG GA G CGA GAT	GGA GGC ATT GCT GAT GAT
*ALP*	NM_001177520.3	GTA TGA GAG TG A CGA GAA	AAT AGG TAG TCC ACA TTG T
*RUNX2*	NM_001024630	AGC AGC ACT CC A TAT CTC T	TTC CAT CAG CGT CAA CAC
*SPP1*	NM_001251830.1	ACG CCG ACC AA G GAA AAC TC	GTC CAT AAA CCA CAC TAT CAC CTC G
*BGLAP*	NM_199173.5	GCA GAG TCC AG C AAA GGT G	CCA GCC ATT GAT ACA GGT AGC
*SP7*	NM_001173467.3	CAG GCT ATG CT A ATG ATT ACC	GGC AGA CAG TCA GAA GAG
*COL1A2*	NM_000089.3	GTG GCA GTG AT G GAA GTG	CAC CAG TAA GGC CGT TTG
*COL2A1*	XM_017018831.3	AGC AGC AAG AGC AAG GAG AA	GTA GGA AGG TCA TCT GGA
*NOG*	NM_001078309.1	CAG CAC TAT CT C CAC ATC CG	CAG CAG CGT CTC GTT CAG
*GREM1*	NM_001191322.1	GAG AAG ACG AC G AGA GTA AGG AA	CCA ACC AGT AGC AGA TGA ACA G
*CHRD*	XM_017007394.1	GCC TCC GCT TCT CTA TCT	AAC AGG ACA CTG CCA TTG

*18S*, reference gene 18S; *GAPDH*, reference gene glyceraldehyde-3-phosphate dehydrogenase; *ALP*, alkaline phosphatase; *RUNX2*, runt-related transcription factor 2; *SPP1*, osteopontin, secreted Phosphoprotein 1; *BGLAP*, osteocalcin, bone gamma-carboxyglutamate protein; *SP7*, osterix, Sp7 transcription factor; *COL1A2*, collagen 1 A2; *COL2A1*, collagen 2 A1; *GREM1*, gremlin 1; *NOG*, noggin; *CHRD*, chordin.

## Data Availability

All the data used in this research are available on request from the first and corresponding author.

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
