# Peer review of "Conditioned Medium of Intervertebral Disc Cells Inhibits Osteo-Genesis on Autologous Bone-Marrow-Derived Mesenchymal Stromal Cells and Osteoblasts"

_biomedicines, 2024, doi:10.3390/biomedicines12020376_

Round 1

Reviewer 1 Report

Comments and Suggestions for Authors

Very interesting manuscript and well organized studies. some questions might need to be addressed to improve the understanding. 

Introduction

Line 74: “the evidence is still not sufficient," please specify what evidence is lacking and briefly describe/summarize the existing gaps in knowledge.

Line 76: CM has been introduced in the part. Please add brief explanation about the CM, how it is relevant to the study and its potential implications in the context of intervertebral disc cells.

Line 85: Please add more explicitly the research gap or question that the study aims to address.

Results

3.1 Line 200-214: Are there specific factors or components in the AF-derived CM that may differ from CEP and NP? are there any discrepancies or unexpected findings (ALP and Alizarin) that require further investigation?

3.2 : The study observes varying degrees of reduced expression in ALP, RUNX2, and SP7 in MSC upon exposure to CM. Are there specific factors in CM from CEP, AF, or NP cells that contribute to the differential effects on these genes? Could the study speculate on potential mechanisms underlying these variations? The observation that other bone-related genes (BGLAP, SPP1, COL1A2) did not show significant reduction in expression compared to the positive control is intriguing. What could be the underlying factors or mechanisms that render these genes resistant to the interference represented by CM? How do the observed changes in gene expression align with potential clinical outcomes, particularly in the context of spinal surgeries? Could the study speculate on the translational implications of these findings for patients undergoing such procedures?

Discussion

Please summarize/highlight precisely how the molecules in CM may interact with different cell types and impact their behavior.

Author Response

                   The reply to reviewer

Reviewer 1

The parts in red are modifications made in response to your questions or suggestions.

Very interesting manuscript and well organized studies. some questions might need to be addressed to improve the understanding. 

Reply to reviewer: We thank the reviewer for the appreciation.  

Introduction

  1. Line 74: “the evidence is still not sufficient," please specify what evidence is lacking and briefly describe/summarize the existing gaps in knowledge.

After modification

We have changed this part on page 2 line 68:

“However, in previous studies all evidence was strictly collected from allogenic donor design, which can have an impact on the degree of inhibition or could be overlaid by other paracrine effects. There is a lack of research on the connection between various cells sources in hIVD and autologous OB or MSC, which are closely related to spinal fusion after discectomy.”

  1. Line 76: CM has been introduced in the part. Please add brief explanation about the CM, how it is relevant to the study and its potential implications in the context of intervertebral disc cells.

Reply to reviewer

In fact, the original text of this paragraph is a brief introduction to CM. As for how it is relevant to the study and its potential implications in the context of intervertebral disc cells, there are few literature reports on the effects of hIVD-derived CM on autologous surrounding cells, such as OBs and MSCs. This is an important reason for conducting this study.

After modification

We have changed this part on page 2 line 78:

“In cell biology and tissue engineering, medium collected from cells is known as the “conditioned medium (CM)” or “secretome” and has gained significant attention in recent years. CM refers to the cell culture medium that has been in contact with primary cells for a specific duration. This medium becomes enriched with various soluble secreted factors, including growth factors, cytokines, and extracellular vesicles, which are released by the cultured cells. These soluble factors can profoundly influence the behavior of cells, including the proliferation and differentiation of cells them-selves [12]. Several studies revealed its role in promoting tissue regeneration [13,14]. Additionally, researchers have also found that animal-derived nucleus pulposus cell (NPC) CM can stimulate heterologous MSC differentiation [15]. However, there are no literature reports on the effects of hIVD-derived CM on autologous surrounding cells, such as OB and MSC.”

  1. Line 85: Please add more explicitly the research gap or question that the study aims to address.

 Reply to reviewer

Thanks for the suggestion. We have revised the introduction so the readers can better understand the problem that this study addresses.

After modification

We have changed this part from page 2 line 68 to the end of introduction:

“…However, in previous studies all evidence was strictly collected from allogenic donor design, which can have an impact on the degree of inhibition or could be overlaid by other paracrine effects. There is a lack of research on the connection between various cells sources in hIVD and autologous OB or MSC, which are closely related to spinal fusion after discectomy.

In cell biology and tissue engineering, medium collected from cells is known as the “conditioned medium (CM)” or “secretome” and has gained significant attention in recent years. CM refers to the cell culture medium that has been in contact with primary cells for a specific duration. This medium becomes enriched with various soluble secreted factors, including growth factors, cytokines, and extracellular vesicles, which are released by the cultured cells. These soluble factors can profoundly influence the behavior of cells, including the proliferation and differentiation of cells them-selves [12]. Several studies revealed its role in promoting tissue regeneration [13,14]. Additionally, researchers have also found that animal-derived nucleus pulposus cell (NPC) CM can stimulate heterologous MSC differentiation [15]. However, there are no literature reports on the effects of hIVD-derived CM on autologous surrounding cells, such as OB and MSC.”

This study aims to investigate the intricate relationship between CM from hIVD and donor-matched OB/MSC, exploring how the factors present in CM could influence cellular behavior of autologous OB/MSC and search for new treatment strategies to improve spinal fusion.”

Results

  1. 1 Line 200-214: Are there specific factors or components in the AF-derived CM that may differ from CEP and NP? Are there any discrepancies or unexpected findings (ALP and Alizarin) that require further investigation?

Reply to reviewer

Unfortunately, we have not conducted an in-depth research on whether there are obvious differences between CM derived from AF and those derived from CEP or NP. But we plan to run proteomic analysis of serum-free medium versus CM of all three IVD cell types in the future. In fact, when AF-derived CM is added to MSC culture, the average expression level of ALP decreases, but there is no statistical difference. The possible reason is that the number of donors is too small (N=3), and another reason is that the variance is too large. These factors all affect AF group results (Figure 3A).

As for why the results of ALP assay on Day 10 in the AF group were different from the ALZR results on Day 21, it may be that as the culture time increases, the inhibitory effect of AF-derived CM on MSC osteogenisis gradually appears (Figure 3A&B). 

  1. 2: The study observes varying degrees of reduced expression in ALP, RUNX2, and SP7 in MSC upon exposure to CM. Are there specific factors in CM from CEP, AF, or NP cells that contribute to the differential effects on these genes? Could the study speculate on potential mechanisms underlying these variations? The observation that other bone-related genes (BGLAP, SPP1, COL1A2) did not show significant reduction in expression compared to the positive control is intriguing. What could be the underlying factors or mechanisms that render these genes resistant to the interference represented by CM? How do the observed changes in gene expression align with potential clinical outcomes, particularly in the context of spinal surgeries? Could the study speculate on the translational implications of these findings for patients undergoing such procedures?

 Reply to reviewer

In this study, when autologous disc-derived CM was added to MSC, the detected bone-related genes showed different change characteristics. The relevant reasons have been mentioned in the discussion section (Changes in these bone-related genes take different times when stimulated). As for the expression in ALP, RUNX2, and SP7 in MSC, we do not know the reason why only some groups have changes in expression but not all. But one thing to note is that the variances of groups with no statistical difference are very large. Different donors (N=3) have greatly different expressions. Perhaps, this is the possible reason.

Regarding the clinical implications of the study, sufficient recovery time needs to be given when evaluating the effect of intervertebral fusion after spinal surgery. Early relevant indicators may not fully reflect the long-term postoperative healing.

This study can explain to a certain extent the possible underlying reasons for some cases of spinal fusion failure, which is due to the presence of residual intervertebral disc tissue during surgery, and these tissues secrete some factors to surrounding osteoblasts and MSCs, thus affecting their normal osteogenesis.

(Related content has been updated in the discussion part from page 11 line 343 to line 351)

Discussion

  1. Please summarize/highlight precisely how the molecules in CM may interact with different cell types and impact their behavior.

After modification

We have changed this part on page 11 line 361:

“Noggin is a well-established antagonist of BMP, which is crucial signaling molecules involved in the regulation of various cellular processes, including differentiation and tissue development [38]. The higher expression of Noggin in AFC, CEPC, and NPC may indicate a potential mechanism that Noggin is secreted into CM by these cell types and inhibits BMP signaling by binding directly to BMP-2, preventing it from interacting with its cell surface receptors. This interference blocks the downstream signaling cascade that leads autologous MSC and OB to osteogenic differentiation, there-by exerting control over their differentiation and functional properties. The findings demonstrate a specific regulatory role of Noggin derived from IVD cells in the cellular microenvironments nearby. So the high expression level of Noggin in human IVD cells may be a potential inhibitor of autologous OB and MSC.”

Reviewer 2 Report

Comments and Suggestions for Authors

The scientific paper "Conditioned medium of intervertebral disc cells inhibits osteogenesis on autologous bone-marrow-derived mesenchymal stromal cells and osteoblasts" aimed to investigate the hypothesis that human intervertebral discs - IVD cells secrete BMP inhibitors that inhibit osteogenesis in autologous osteoblast (hOB) and bone marrow mesenchymal stem cells (hMSC). It can be considered that:

1)      Remove the titles from the abstract as "the abstract should be a single paragraph and should follow the style of structured abstracts, but without headings" (instructions for authors);

2)      The abstract should be more concise. Maximum 250 words. I suggest reducing the background;

3)      Rewrite the introduction to lines 81-84;

4)      In the introduction, before the objective, insert a sentence with the originality of the manuscript (Gap in the literature) that justifies the research;

5)      Insert the ethical aspects of the research into the methodology. What was the approval opinion number, date and responsible committee;

6)      Make it clear in the methodology how many patients participated in the research, the eligibility criteria and location of recruitment;

7)      Insert the histology results into the manuscript, with images;

8)      Review the discussion as the results could be better discussed in full, including the histology that was not presented;

9)      Insert the limitations of the study at the end of the discussion;

10)  Include the clinical potential of the study in the conclusions;

11)  Insert at least 10 more references, from the last 3 years. The number of references is low and outdated.

Comments on the Quality of English Language

Moderate editing

Author Response

                 The reply to reviewer

Reviewer 2

The parts in red are modifications made in response to your questions or suggestions.

The scientific paper "Conditioned medium of intervertebral disc cells inhibits osteogenesis on autologous bone-marrow-derived mesenchymal stromal cells and osteoblasts" aimed to investigate the hypothesis that human intervertebral discs - IVD cells secrete BMP inhibitors that inhibit osteogenesis in autologous osteoblast (hOB) and bone marrow mesenchymal stem cells (hMSC). It can be considered that:

1)      Remove the titles from the abstract as "the abstract should be a single paragraph and should follow the style of structured abstracts, but without headings" (instructions for authors).

2)      The abstract should be more concise. Maximum 250 words. I suggest reducing the background.

Reply to reviewer

Thanks for your suggestions. We have revised the abstract according to the two comments above.

3)      Rewrite the introduction to lines 81-84.

4)      In the introduction, before the objective, insert a sentence with the originality of the manuscript (Gap in the literature) that justifies the research.

Reply to reviewer

Thanks for your suggestions. We have revised the introduction according to the two comments above. They are updated as follows (page 2 line 78):

After modification

In cell biology and tissue engineering, medium collected from cells is known as the “conditioned medium (CM)” or “secretome” and has gained significant attention in recent years. CM refers to the cell culture medium that has been in contact with primary cells for a specific duration. This medium becomes enriched with various soluble secreted factors, including growth factors, cytokines, and extracellular vesicles, which are released by the cultured cells. These soluble factors can profoundly influence the behavior of cells, including the proliferation and differentiation of cells them-selves [12]. Several studies revealed its role in promoting tissue regeneration [13,14]. Additionally, researchers have also found that animal-derived nucleus pulposus cell (NPC) CM can stimulate heterologous MSC differentiation [15]. However, there are no literature reports on the effects of hIVD-derived CM on autologous surrounding cells, such as OB and MSC.

This study aimed to investigate the relationship between CM of hIVD and donor-matched OB/MSC, exploring how the factors present in CM could influence cellular behavior of autologous OB/MSC and search for new treatment strategies to improve spinal fusion.

5)      Insert the ethical aspects of the research into the methodology. What was the approval opinion number, date and responsible committee.

6)      Make it clear in the methodology how many patients participated in the research, the eligibility criteria and location of recruitment.

Reply to reviewer

The two questions above are replied together. We communicated with the journal about this when submitting the article. This is the content of the communication. We will make relevant changes in this regard, thank you for your suggestions. The previous email is as below.

“Dear Ms Chankong,

thanks for reaching out concerning the ethics.

We have a “general consent” in place at the University Hospital that allows us to use human materials form spinal surgeries:

"Bone marrow aspirates were obtained from patients undergoing spine surgery at the Inselspital Bern. The Swiss Human Research Act does not apply to research that involves anonymized biological material and/or anonymously collected or anonymized health-related data. Therefore, this project did not need to be approved by the ethics committee. General Consent which also covers anonymization of health-related data and biological material was obtained.”

On top of that we have the following ethical approval in place specifically to use human IVD tissues and bone-marrow samples:

      > Ethic Committee Name: Swissethics

      > Approval Code:SwissEthics # 2019-00097

      > Approval Date:2 May 2019

Best”

After modification

We have changed this part on page 2 line 91:

2.1. Human materials and cell isolation

Bone fragments, intervertebral disc tissues and human bone marrow aspirates were collected from same patients as an inclusion criterion undergoing spinal surgery in Insel hospital (Table 1, 2). Seven donors derived MSC/OB cells were for culture with CM from autologous IVD and nine donors for comparison of Noggin expression be-tween IVD and autologous MSC/OB. All the donor tissues and cells were anonymously collected with written consent. The approvals were either under the Swiss-Ethics approval (# 2019-00097) or were obtained under the general consent of the Insel hospital, which also covers anonymization of health-related data and biological material.

7)      Insert the histology results into the manuscript, with images;

Reply to reviewer

We have already presented the histological results in the previous version in Figure 3B&D. The cells cultured with CM were harvested and fixed on Day 21, followed by alizarin red staining to detect the mineralized matrix in human OBs/MSCs. Maybe the pictures are small, and it is hard to read more details. Now we have now included a supplementary online Figure S2 where high-resolution images at higher magnification can be obtained.

8)      Review the discussion as the results could be better discussed in full, including the histology that was not presented;

Reply to reviewer

The discussion section has been modified accordingly in the new version on page 10 line 319. As for histology, it has been mentioned in passage 3 of the discussion section in the original version.

9)      Insert the limitations of the study at the end of the discussion.

Reply to reviewer

Thanks for the suggestion. Now the limitations of the study have been modified in the new version.

After modification

We have changed this part on page 11 line 370:

“The findings demonstrate a specific regulatory role of Noggin derived from IVD cells in the cellular microenvironments nearby. So the high expression level of Noggin in human IVD cells may be a potential inhibitor of autologous OB and MSC.

However, our research has some limitations. Noggin concentration in CM was not measured at protein level at this time. Proteomics of the CM could be performed in the future to investigate the precise and possible influence from other components in CM on the experimental results.”

10)  Include the clinical potential of the study in the conclusions.

Reply to reviewer

The conclusions are updated in a new version about the clinical potential on page 11.

After modification

Our study reveals a distinctive difference of high Noggin expression in human AFC, CEPC, and NPC relative to of autologous OB and MSC. The findings also demonstrate the involvement of human IVD cells derived molecules, including Noggin potentially, in the regulation of autologous OB and MSC and the inhibition of osteogenesis in these cells by paracrine potentially. It offers an insight into the process of osteogenesis and knockdown of Noggin in human IVD cells, perhaps, improve the postoperative spinal fusion. While these findings present promising avenues for therapeutic interventions. Further research is needed to fully understand the potential regulatory mechanism. Importantly, the complete removal of the affected IVD during spinal surgery for LBP can effectively block the impact of IVD on osteogenesis, which plays a crucial role in postoperative intervertebral fusion.

11)  Insert at least 10 more references, from the last 3 years. The number of references is low and outdated.

Reply to reviewer

Thanks for your suggestions. They are updated. About 50% are from the last 3 years.

Reviewer 3 Report

Comments and Suggestions for Authors

The study aimed to investigate the reason for postoperative osteogenesis inhibition by focusing on the role of IVD cells and secreted BMP inhibitors that inhibit osteogenesis in autologous osteoblasts and MSCs. However, there are serious issues in this study as the following:

(1) No evidence that isolated cells are hOB or MSCs. You have to characterize the isolated cells and provide evidence that isolated cells represent the morphologic and genetic nature of the target cells.  

(2) In this study, no direct evidence exists that Noggin mRNA delivered to hOB or MSCs and showed osteogenesis impairment. And how that could happen?

(3) Osteogenesis stimulation and inhibition should be tested in vivo using mice because always in vitro experiments don't reflect the same environment in vivo. the conditions are very different.  You have to show some in vivo evidence supporting your conclusion. 

(4) how much volume was the culture medium hOB and MSCs and how much volume did you inject from IVD cell culture? Have you tested different rations? is the culture of IVD cells old or 3 days?  did you test different culture timing, and what are your standards? 

(5) Have you tried to recover cells after treatment by changing the medium with a new medium containing stimulation factors to see reverse effects? 

I think this study needs a lot of work to come up with this conclusion. 

Comments on the Quality of English Language

No comments. 

Author Response

                 The reply to reviewer

Reviewer 3
The parts in red are modifications made in response to your questions or suggestions.

The study aimed to investigate the reason for postoperative osteogenesis inhibition by focusing on the role of IVD cells and secreted BMP inhibitors that inhibit osteogenesis in autologous osteoblasts and MSCs. However, there are serious issues in this study as the following:
(1)    No evidence that isolated cells are hOB or MSCs. You have to characterize the isolated cells and provide evidence that isolated cells represent the morphologic and genetic nature of the target cells.  
Reply to reviewer
We have routinely tested the isolated cells for the expression of major ECM genes, which are highly distinguishable among the tissues. MSCs were routinely checked for CD marker expression as defined by Dominici et al. [1], see Figure 1 below from flow cytometry for second passage MSCs. However, the publication by Whitney et al. [2] has concluded that there are no good CD markers for MSCs or musculoskeletal cells in general. This study clearly proved that donor-matched skin fibroblasts were showing the same CD marker profiles. Thus, trilineage differentiation into osteocytes, adipocytes and chondrocytes is the only way MSCs can be clearly identified and characterized. However, as we did not do trilineage of our MSCs here but only osteogenic because of the specific study design, we kept the label of “stromal cells” as we did not prove multipotency here.

Thus, to conclude CD marker expression are not very helpful. However, we have now performed additional qPCR analysis and have analyzed prominent ECM genes of our primary cells after low passage expansion. We have chosen the col2/col1 ratio, which is an indicator for the origin of these cell types. We provide this figure as supplementary Figure S1.
Supplemenary online Figure S1: Col2/Col1 gene expression ratio of donor-matched primary cell types at passage 2-4 in OBs, MSCs, AFC, CEPC, and NPC. The graph clearly demonstrates that AFC, CEPC and NPC in increasing order do express higher levels of col2, which is absent in OBs and MSCs. Mean ± SD, N = 3 donors. 

(2)    In this study, no direct evidence exists that Noggin mRNA delivered to hOB or MSCs and showed osteogenesis impairment. And how that could happen?
Reply to reviewer
From previous research in our group, we could show that osteogenesis of OB can be induced with BMP2 and also can be impaired by Noggin in a dose-dependent way [3]. In the updated manuscript version, we have added a limitation on research at the end of the discussion on page 11 line 370. We are currently performing new experiments where we will focus on the composition of the CM and try to find out the most important factors that are responsible for this inhibition. This is very interesting scientifically. Clinically, we already have identified a BMP2 analog (L51P), which generally inhibits obviously a wider range of BMP antagonists. This will be addressed in a future study.  
(3)    Osteogenesis stimulation and inhibition should be tested in vivo using mice because always in vitro experiments don't reflect the same environment in vivo. The conditions are very different.  You have to show some in vivo evidence supporting your conclusion. 
Reply to reviewer
Indeed, animal experiments can further confirm the results of the study. However, animal study is out of scope for this study where the primary aim was to demonstrate inhibition in autologous donor samples and not in an allogenic design.
As what concerns the animal experimentation, we are in progress of addressing exactly this issue of the lacking immune response in a rat animal model for improved spinal fusion. Spinal fusion is possibly impaired by the presence of a high level of antagonists in the IVD niche. Thus, spinal fusion is a challenge, and the main reason that only very high doses of BMP2 must be used to achieve sufficient results [4,5].  We have now established a protocol to simulate discectomy in an elderly rat animal model [6]. We also successfully show that certain mixtures using BMP2 and BMP2 analogues, such as the L51P [7,8], (that is not in the focus of this present study) do have a very potent effect on improved ossification [9]. In the future we plan to test by specific inhibition of Noggin, gremlin 1 or chordin (work in progress)

(4)    how much volume was the culture medium hOB and MSCs and how much volume did you inject from IVD cell culture? Have you tested different rations? is the culture of IVD cells old or 3 days?  did you test different culture timing, and what are your standards? 
Reply to reviewer
In brief, put fresh medium (4ml) into IVD cells (30 beads in each well, about 106 cells/bead) to culture for 3 days and collect the CM. Then together with osteogenic medium (ratio, CM : osteogenic medium = 1:1) and added 2ml to MSC/OB.
We did not test different ratios. This would be a very interesting study design in the future. However, the production of the CM was time-consuming, and the timing of the four different primary cell cultures was challenging. Thus, we decided to stick to 1:1 ratio in this current design.
Regarding the culture time, we have actually tested the culture for 11 days, 14 days and 21 days. In previously published studies in our laboratory, the optimal harvesting time point for ALP assay was on Day 11, while alizarin red staining was performed on Day 21. This is an important reason why we set these two time points. 
Regarding the details of the acquisition and cultivation of CM, we have already explained it in the Materials and Methods section of the original article. The details are as follows (from manuscript):
“2.2. Alginate bead and conditioned medium
For generation of the conditioned medium, human OB, MSC and IVD cells (AFC, CEPC and NPC; passage 2-4) were encapsulated in 1.2% alginate (Fluka, Basel, Switzerland) dissolved in 0.9% sodium chloride solution at a density of 4 million/ml. Then the alginate solution was flown at a constant rate at 1.5 ml/min (syringe pump TI - Part # 78-8100 - Model No. 100, KD Scientific, Holliston, US) through a 22G needle and dropped into a 102 nM CaCl2 salt solution, which immediately solidified and generated the beads. Within 5 min, the beads underwent rinsing with a 0.9% sodium chloride solution before being transferred to 6-well plates (30 beads/well) and cultured with 4ml α-MEM (10% FBS, 1% P/S). After three days, the CM was collected and filtered with a 0.22μm filter for use (Figure 1).  
2.3. Cell culture with conditioned medium
Human OB/MSC cells were seeded at 2 × 104 cells/well in 12-well plates in the ba-sal medium (α-MEM; 10% FBS, 1% P/S) and left overnight for cell adherence. Osteogenic medium (α-MEM containing 10% FBS, 1% P/S, 0.1 mM L-ascorbic acid-2-phosphate, 20 mM beta-glycerophosphate and 200 nM Dexamethasone, from Sigma-Aldrich), together with conditioned medium from corresponding cells, were added into each well (2ml, ratio 1:1), including positive control, AFC group, CEPC group and NPC group. Negative control was only added the basal medium. The medium was refreshed twice a week (Figure 2).”

(5)    Have you tried to recover cells after treatment by changing the medium with a new medium containing stimulation factors to see reverse effects? 
Reply to reviewer
This is a very good point, but we have not yet mastered the timepoint to join a new medium containing stimulation factors. Additionally, after the cells were induced (also together with CM) for 21 days, because cells have been over-confluent for a long time in 24-well plates, if we add stimulation factors inside to observe the possible results, it will take another 21 days. Maybe the cells in the well are died because they have been cultured for 42 days and fully grow for over 30 days (usually the cells in the well will be full on Day 10)

I think this study needs a lot of work to come up with this conclusion. 
 Reply to reviewer

Thanks for your comments. The conclusion has been revised accordingly.

References

1.    Dominici, M.; Le Blanc, K.; Mueller, I.; Slaper-Cortenbach, I.; Marini, F.C.; Krause, D.S.; Deans, R.J.; Keating, A.; Prockop, D.J.; Horwitz, E.M. Minimal criteria for defining multipotent mesenchymal stromal cells. The International Society for Cellular Therapy position statement. Cytotherapy 2006, 8, 315-317, doi:https://doi.org/10.1080/14653240600855905.
2.    Whitney, M.J.; Lee, A.; Ylostalo, J.; Zeitouni, S.; Tucker, A.; Gregory, C.A. Leukemia Inhibitory Factor Secretion is a Predictor and Indicator of Early Progenitor Status in Adult Bone Marrow Stromal Cells. Tissue Engineering Part A 2008, 15, 33-44, doi:10.1089/ten.tea.2007.0266.
3.    Albers, C.E.; Hofstetter, W.; Sebald, H.-J.; Sebald, W.; Siebenrock, K.A.; Klenke, F.M. L51P — A BMP2 variant with osteoinductive activity via inhibition of Noggin. Bone 2012, 51, 401-406, doi:https://doi.org/10.1016/j.bone.2012.06.020.
4.    Tannoury, C.A.; An, H.S. Complications with the use of bone morphogenetic protein 2 (BMP-2) in spine surgery. The Spine Journal 2014, 14, 552-559, doi:https://doi.org/10.1016/j.spinee.2013.08.060.
5.    Simmonds, M.C.; Brown, J.V.E.; Heirs, M.K.; Higgins, J.P.T.; Mannion, R.J.; Rodgers, M.A.; Stewart, L.A. Safety and Effectiveness of Recombinant Human Bone Morphogenetic Protein-2 for Spinal Fusion. Annals of Internal Medicine 2013, 158, 877-889, doi:10.7326/0003-4819-158-12-201306180-00005.
6.    Oswald, K.A.C.; Bigdon, S.F.; Croft, A.S.; Bermudez-Lekerika, P.; Bergadano, A.; Gantenbein, B.; Albers, C.E. Establishment of a Novel Method for Spinal Discectomy Surgery in Elderly Rats in an In Vivo Spinal Fusion Model. Methods and Protocols 2021, 4, doi:10.3390/mps4040079.
7.    May, R.D.; Frauchiger, D.A.; Albers, C.E.; Hofstetter, W.; Gantenbein, B. Exogenous Stimulation of Human Intervertebral Disc Cells in 3-Dimensional Alginate Bead Culture With BMP2 and L51P: Cytocompatibility and Effects on Cell Phenotype. Neurospine 2020, 17, 77-87, doi:10.14245/ns.2040002.001.
8.    Hauser, M.; Siegrist, M.; Denzer, A.; Saulacic, N.; Grosjean, J.; Bohner, M.; Hofstetter, W. Bisphosphonates reduce biomaterial turnover in healing of critical-size rat femoral defects. Journal of Orthopaedic Surgery 2018, 26, 2309499018802487, doi:10.1177/2309499018802487.
9.    Gantenbein B, O.K., Erbach GF, Croft AS, Bermudez Lekerika P, Strunz F, Bigdon S, Albers CE. The Bone Morphogenetic Protein 2 Analogue L51P Enhances Spinal Fusion in Combination with BMP2 in an In Vivo Rat Tail Model (in revision). Acta Biomater 2024.

Round 2

Reviewer 2 Report

Comments and Suggestions for Authors

No comments 

Comments on the Quality of English Language

Minor editing

Author Response

We thank the reviewer for the comments. 

Reviewer 3 Report

Comments and Suggestions for Authors

The author's responses were mostly acceptable. But, I suggest elaborating on the unperformed experiments in the study limitations. 

Comments on the Quality of English Language

I think mostly okay. 

Author Response

We followed the two reviewer’s suggestions “elaborating on the unperformed experiments in the study limitations” and made final edits. As the first author is not a native English speaker we gave the manuscript for editing to a native English speaker, which took some extra time. Furthermore, we have worked hard on an improved discussion and also more recent and updated references on the subject.